# Investigation of the Flow Characteristics for Cylinder-in-Ball Valve Due to a Change in the Opening Rate

Hyo-Lim Kang, Hyung-Joon Park and Seung-Ho Han *

A Mechanical Engineering, Dong-A University, Busan 49315, Korea
* Correspondence: shhan85@dau.ac.kr; Tel.: +82-51-200-7655

**Abstract:** Ball valves are widely used as flow control devices wherein a hollow, perforated ball pivots to control the liquid flowing through it. Due to a simple structure and only a quarter-turn of the stem required for either a fully open or fully closed state, ball valves are known for being durable with excellent shut-off properties, albeit without offering precise flow control. V-port ball valves are used as an alternative to allow for linear and even equal percentage flow characteristics, but a robust construction is required for higher velocity working fluids in the small end opening of the V-shape; otherwise, the valve could sustain damage. In the present study, a cylinder-in-ball valve is proposed through a structural conceptual design, wherein the opening begins at the center of the flow path. The flow characteristics and flow rates according to the opening rate were quantitatively evaluated via computational fluid dynamic analysis. The results showed that the flow coefficient, CV, with a range of 1.05~109.87, increased exponentially over the opening rates of 20~100%. A numerical analysis for the multi-phase flow was performed to calculate the vapor volume fraction to confirm the effects of cavitation. In addition, an experiment was conducted on the CV values to verify the validity of applying the proposed cylinder-in-ball valve as a flow control valve. Good agreement for the CV values was obtained between the experimental and numerical results.

**Keywords:** control valve; cylinder-in-ball valve; flow coefficient; inherent flow characteristics; structural conceptual design

## 1. Introduction

Control valves that provide precise flow control would be a major component in fluid transportation. Precise flow control plays an important role in process industries such as nuclear power plants, petrochemical plants, and thermal power plants [1]. Among them, ball valves are widely used as flow control devices. The conventional ball valve design involves a hollow, perforated, or pivoting ball to control the flow of working fluids [2]. Due to a simple structure that requires only a quarter-turn of the stem for a fully open or fully closed state, ball valves are known for their durability and excellent shut-off properties. This design, however, does not always allow for precise flow control as a result of unique flow characteristics [3].

Reports from related studies have described the phenomenon of these flow characteristics. Thananchai [4] conducted a study focused on improving the flow characteristics of the ball valve. The study showed that irregular flow occurred due to changes in the flow path caused by the opening rate. The irregular flow rate was influenced by the shape and size of the opening hole in the ball. Those factors were improved by altering the placement of the existing hole to prevent complications with the flow path. Tabrizi et al. [5] carried out computational fluid dynamic and experimental analysis to investigate the flow characteristics of the ball valve that were due to changes in the opening rate. As the opening rates became smaller, complicated flow paths gradually increased the presence of vortices in the valve trim. The vortex-induced cavities were found to be a major hindrance to precise flow control. Chern et al. [6] experimentally studied the flow coefficients and cavitation in ball

valves. The pressure loss coefficient, K, and the flow coefficient, *CV*, were estimated by measuring the pressure and flow rate via steady-flow testing. Cavitation was evaluated via a cavitation index that was determined through pressure data measurements. The authors found that vortices in the valve trim were the main cause of the cavitation that was significantly influencing the flow characteristics.

The studies mentioned above have shown that significant vortices occur with small opening rates, which result in irregular flow characteristics. Vortices complicate precise flow control in the ball valve. They are a major contributor to cavitation and, in addition, exacerbate safety problems in piping systems, so that the application of a flow control valve to the ball valve has limitations. To overcome these limitations, researchers have improved the shape of the opening hole in the ball and given it a V-shaped geometry, which is the so-called V-port ball valve.

Wang et al. [7] proposed a V-port ball valve with V-shaped angles of 30, 60, and 90 degrees and investigated the resulting flow characteristics. The results showed that the V-port ball valve improved the irregular flow characteristics of the conventional ball valve. Decreases in the V-shaped angles, however, resulted in a loss of pressure and a remarkable occurrence of the cavitation phenomenon. With a V-shaped angle of 30 degrees, cavitation occurred even when the valve was fully opened. In a similar study, Gao et al. [8] proposed an improved V-port ball valve with two types of ports, i.e., approximately triangular and rectangular ports, and investigated the flow characteristics for each port type. The *CV* values for both types increased as the opening rates of the valve increased. Under the same opening rate, however, the V-port ball valve with an approximately rectangular port showed higher *CV* values. Merati et al. [9] used Strouhal and Reynolds numbers to study the flow characteristics of the V-port ball valve. These numbers helped to construct a five-dimensional polynomial relationship that explained the occurrence of a dominant vortex at the end of the valve trim. To minimize the vortex, the authors proposed an increase in the pipe diameter, a restriction in the flow rate, and the addition of an orifice downstream.

The V-port ball valve has helped overcome the limitations of using the conventional ball valve to control the irregular flow characteristics. However, the occurrence of excessive vortices and cavitation persists. In the present study, a cylinder-in-ball valve is proposed wherein the opening hole begins at the center of the flow path when a device is used to implement the counter-rotating motion of the cylinder and ball. Computational fluid dynamic analysis was carried out to quantitatively evaluate the *CV* when changing the opening rate in the flow path. Then, we evaluated whether the proposed cylinder-in-ball valve met the inherent flow characteristics requirements, such as equal percentage, that are required for precise flow control. Subsequently, an experiment using flow test equipment was performed, and the validity of using the proposed cylinder-in-ball valve for flow control was verified.

## 2. Conceptual Design for Cylinder-in-Ball Valve

The conventional ball valve has a light and simple structure compared with other valves. Since the opening starts from the side, however, the internal flow path becomes very complicated. The complicated flow path induces vortices and irregular flow characteristics. The vortices occur due to the complicated flow path, which cause a significant increases in energy loss, noise, and vibration. These lead to an instantaneous decrease in the *CV* values, which is a main causative factor in irregular flow characteristics. In the case of a small opening rate, irregular flow persists even when the opening rate remains the same. [10] The vortices and irregular flow characteristics make it difficult to precisely control the flow rate. The turbulence of the flow generated by the vortices can even cause cavitation. Therefore, it is necessary to improve the complicated flow path. Thus, a structural conceptual design was developed so that the opening would start at the center of the flow path when the valve is opened.

A cylinder-in-ball valve with a pipe diameter of 1-1/2″ is proposed in this study, in which a cylinder was located into a ball, as shown in Figure 1a. The ball and cylinder were

connected to the outer and inner stems, respectively, which rotate in opposite directions via a combination of gears. The inner stem was connected to a spur gear with 20 teeth and a module of 2.25 mm, and the outer stem was connected to a spur gear with the same teeth but a larger module of 4.5 mm. These gears were rotated in opposite directions by an idler of a two-stage gear train, which was linked by the handle to manually control the opening rate. This arrangement of the gear train provided a coaxial counter-rotating motion of the ball and cylinder, which allowed the valve opening to start at the center of the flow path. To support the ball and allow for easy maintenance, a trunnion type and a split body were adopted, respectively. The ball and cylinder had the same size of perforated holes with diameters of 1-1/4", as shown in Figure 1b, where each perforated hole can be identified in the cross-sectional view on the XZ-plane according to the Cartesian coordinates. Figure 1c shows the coaxial counter-rotating motions of the ball and cylinder at an opening rate of 20% in the cross-sectional views on the XY- and XZ-planes, respectively. The flow path opens almost in the middle, while a conventional ball valve begins to open on the side.

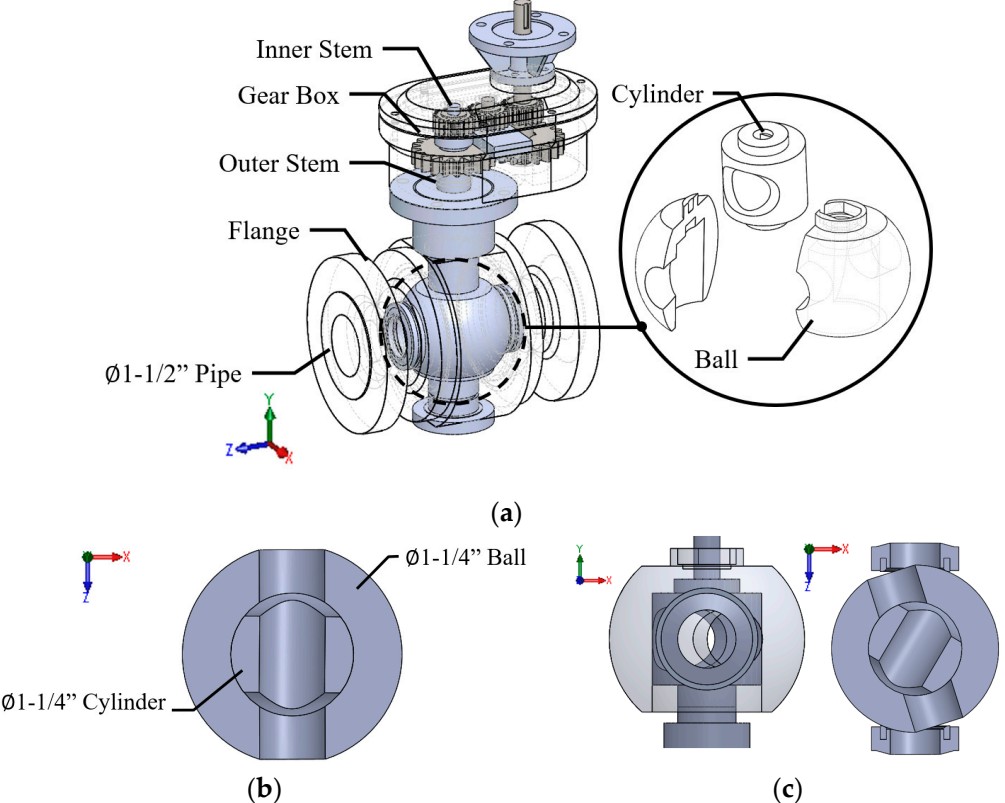

**Figure 1.** Configuration of the proposed cylinder-in-ball valve: (**a**) 3D model of the cylinder-in-ball valve; (**b**) perforated holes in the ball and cylinder; (**c**) coaxial counter-rotating motion of the cylinder and ball at an opening rate of 20%.

From the conceptual design of the cylinder-in-ball valve provided detailed design drawings, and a prototype was manufactured, which used the material SUS 316L for the major components. The prototype was used for experimental *CV* measurements to verify the results of the CFD analysis.

## 3. Inherent Flow Characteristics and Numerical Analysis

### 3.1. Inherent Flow Characteristics

The characteristics that indicate the unique flow rate performance of valves according to the opening rates are referred to as the inherent flow coefficients [11]. The flow characteristics of valves vary according to the inherent flow coefficients, which are significantly affected by valve type. Flow characteristics are an important factor in selecting a control

valve, and these are divided into three types: equal percentage, linear, and quick opening. Figure 2 shows the curves regarding the opening rates of inherent flow characteristics for various types of valves. The equal percentage characteristic describes a flow rate that increases exponentially with increases in the valve opening rate, and this is the most common characteristic of control valves. The linear characteristic refers to a flow rate that constantly increases with increases in the valve opening rate. The quick opening characteristic allows for a significant amount of flow to pass through the initial opening of the valve. Among the three characteristics, flow control valves that use the equal percentage characteristic make up 80% of the control valves used in industry because it shows higher efficiency in controlling major factors such as flow rate and pressure, and this feature provides sophisticated flow control even when a small opening rate is used.

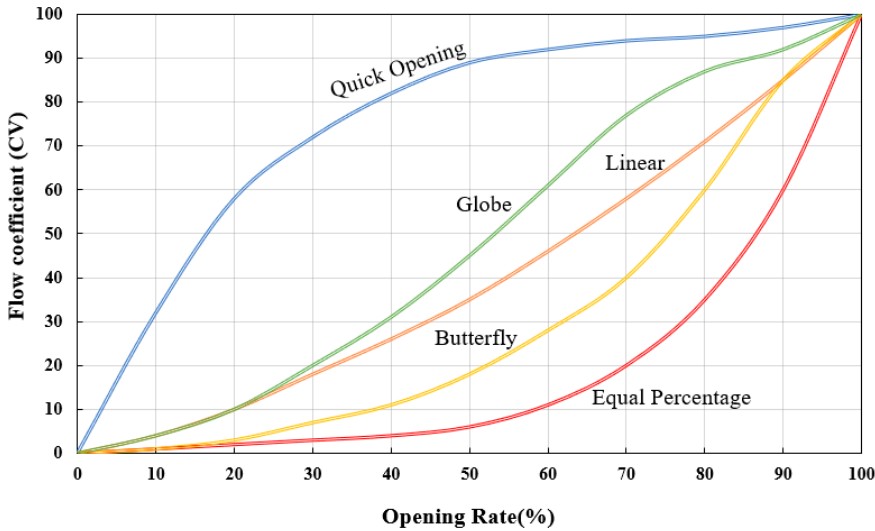

**Figure 2.** The curves of the inherent flow characteristics for various valves.

### 3.2. Flow Coefficients

Based on the conceptual design described in Section 2, the cylinder-in-ball valve was proposed with the opening hole starting at the center of the flow path. In order to evaluate the flow characteristics of the proposed cylinder-in-ball valve, computational fluid dynamic (CFD) analysis was carried out. CFD analyses of the flow coefficient, *CV*, according to the opening rates were derived by calculating the velocity and pressure distributions. These distributions confirmed the inherent flow coefficients of the proposed cylinder-in-ball valve. Then, the cavitation that occurred at small opening rates was evaluated quantitatively and compared with that of a conventional V-port ball valve [12].

To evaluate the flow characteristics, CFD analysis using the commercial software ANSYS 2020 R2 CFX (San Jose, CA, USA) [13] was carried out for each opening rate of the proposed cylinder-in-ball valve, in which the domestic code KS B 2101 [14], standardized for *CV* measurement, was taken into account. This code defined the working fluid as water at room temperature and stipulated that the upstream length from the inlet to the valve and the downstream length from the valve to the outlet should be 6 and 10 times longer than the pipe diameter, respectively. For the pre-processing of the CFD analysis, the finite element model was prepared, as shown in Figure 3. To construct the grid of the finite element model, the hex-dominant method was adopted to generate preferentially hexahedral elements in order to provide highly accurate results for the CFD analysis. The average number of elements was approximately 2.5 million, depending on the opening rates.

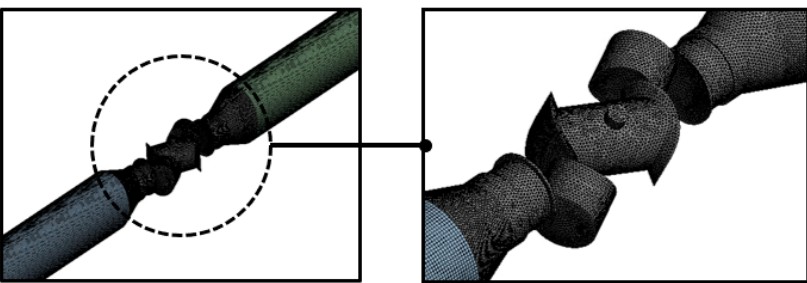

**Figure 3.** Features of the finite element model.

The *CV* was calculated using the *Q* at the outlet obtained by the results of the CFD analysis, as shown in Equation (1).

$$CV = 1.17 \, Q \sqrt{\frac{SG}{\Delta P}} \, \left[\text{gal·min}^{-1}\text{·psi}^{-0.5}\right] \tag{1}$$

$$Q = V{\cdot}A \, \left[\text{m}^3\text{·h}^{-1}\right] \tag{2}$$

In Equation (1), a coefficient of 1.17 was used to convert SI units into I-P units for the practical use of *CV* values. From the above equations, *Q*, *SG*, and *V* are the volumetric flow rate (m$^3$/h) at the outlet, the specific gravity of water (1), and the average velocity (m/s) at the cross-sectional area, *A* (m$^2$), respectively. $\Delta P$ is the differential pressure (kPa) between the inlet and outlet, which is designated as 74 kPa according to KS B 2101. The *Q* at the outlet was determined by the CFD analysis using the boundary conditions such as the *Q* at the inlet and atmospheric pressure at the outlet met the requirement for a $\Delta P$ of 74 kPa.

### 3.3. Cavitation

Complicated flow paths in valves often causes cavitation due to rapid changes in the pressure and velocity of the working fluid. Cavitation occurs when the fluid pressure falls below the saturated vapor pressure, and the vapors that are generated tend to form cavities. The cavities move to an area above the saturated vapor pressure. At that point, they collapse rapidly and create shock waves. These shock waves disturb not only flow control but could even damage the inside of the valve [15].

The vapor volume fraction, *VVF*, has been used to quantify the degree of damage due to cavitation, as shown in Equation (3).

$$VVF = \frac{V_{vapor}}{V_{element}} \tag{3}$$

In Equation (3), $V_{vapor}$ is the volume occupied by vapors, and $V_{element}$ is the volume of one element in the finite element model. The $V_{vapor}$ was calculated using a form of multi-phase CFD analysis that considers both liquid and gaseous phases.

To account for the turbulence characteristics of cavitation at the rear portion of the flow path and the turbulent stress that occurs due to the perturbation of the fluid velocity, the Reynolds-averaged Navier–Stokes (RANS) equation [16] was adopted. For the turbulence model, the SST k-ω model [17] was used, because both flow separation and multi-phase flow at the wall could be implemented. To calculate cavity size, such as the radius of vapors, $R_B$, from the results obtained by the CFD analysis via the RANS equation, the Rayleigh–Plesset equation was applied, as shown in Equation (4). Subsequently, the $V_{vapor}$ was determined by the $R_B$ [18].

$$R_B \frac{d^2 R_B}{dt^2} + \frac{3}{2}\left(\frac{dR_B}{dt}\right)^2 + \frac{2\sigma}{\rho_f R_B} = \frac{p_v - p}{\rho_f} \tag{4}$$

In Equation (4), $\sigma$, $p_v$, $p$, and $\rho_f$ are the surface tension coefficient, saturated vapor pressure (Pa), fluid pressure (Pa), and fluid density (kg/m$^3$), respectively.

## 4. Results

### 4.1. Flow Coefficients

The *CV* regarding a change in the opening rate of the proposed cylinder-in-ball valve was calculated using Equation (1). The CFD analysis was carried out using boundary conditions such that the *Q* at the inlet and atmospheric pressure at the outlet met the requirement for a $\Delta P$ of 74 kPa. The *Q* at the inlet regarding a change in the opening rate of the valve was found using a trial-and-error approach. A series of CFD analyses under pressure boundary conditions, such as 74 kPa at the inlet and atmospheric pressure at the outlet, were repeated until the desired values of the *Q* at the inlet was obtained. Then, the obtained *Q* at the inlet for each opening rate and atmospheric pressure at the outlet were designated as the boundary conditions for the CFD analysis. Subsequently, the final CFD analysis was observed. Table 1 summarizes the boundary conditions for the CFD analysis.

**Table 1.** The boundary conditions of the CFD analysis according to the opening rates.

| | Opening Rate (%) | | | | |
|---|---|---|---|---|---|
| | 20 | 30 | 50 | 75 | 100 |
| Inlet flow rate (m$^3$/h) | 0.71 | 2.10 | 7.92 | 21.6 | 81.09 |
| Outlet pressure (atm) | | | 1 | | |

Figure 4 shows the streamlines of the working fluid in the valve at an opening rate of 20%. As shown in Figure 1c, the opening started at the center of the cross-section in the flow path so that the streamlines approximated a simple straight line. Although several vortices were generated at the region of contact with the ball and cylinder, the vortices were small enough to not significantly affect the main flow path. These did not lead to a sudden change in the local pressure in the flow path.

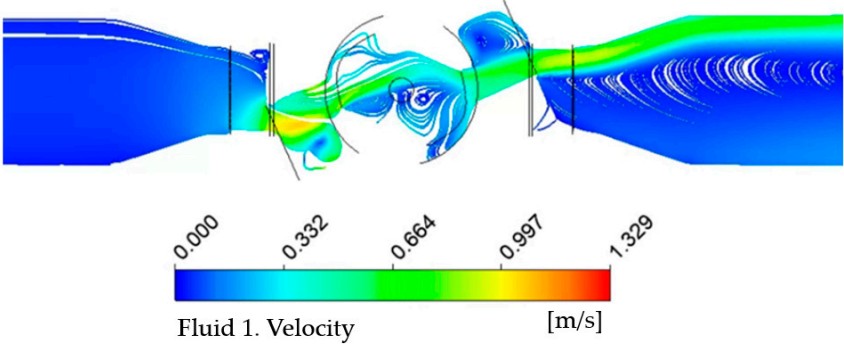

**Figure 4.** Streamlines at an opening rate of 20% for the cylinder-in-ball valve.

The obtained *Q* at the outlet and the *CV* calculated using Equation (1) with regard to changes in the opening rate of the valve are plotted in Figure 5. As the opening rate was changed to 20, 30, 50, 75, and 100%, the *Q* increased to 0.77, 2.20, 8.05, 21.82, and 80.78 m$^3$/h, respectively. The *CV* values increased similarly to the rates for *Q*, such as 1.05, 2.99, 10.95, 29.68, and 109.87. The changes of the *CV* with the opening rate followed in orderly exponential increments.

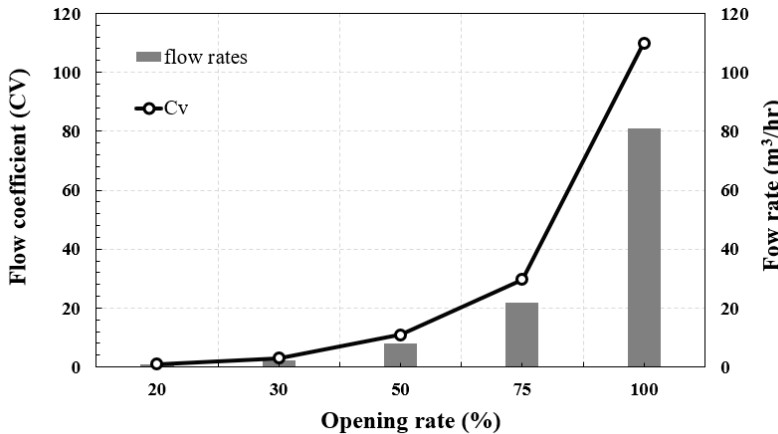

**Figure 5.** *CV* and *Q* values due to changes in the opening rate.

These flow characteristics were compared with those of a conventional ball valve and a commercial V-port ball valve with the same pipe diameter of 1-1/2" and a V-port angle of 90°. Figure 6 shows the *CV* values as the opening rate increases for the proposed cylinder-in-ball valve, a conventional ball valve, and the V-port ball valve. The increments of *CV* values showed very similar behaviors for three types of valves. The conventional ball valve showed higher *CV* values for the entire range of opening rates, but with irregular flow characteristics at some specific regions. The conventional ball valve was difficult to use as a valve for sophisticated flow control. Meanwhile, in the case of the V-port ball valve, no irregular flow characteristics were found for the entire region of opening rates, and the *CV* increased exponentially: 1.07, 4.00, 20.78, 40.05, and 90.50. For the region of opening rates between 40 and 80%, however, the proposed cylinder-in-ball valve provided an insensitive *CV* change compared with that of the V-port ball valve. This suggests that the proposed cylinder-in-ball valve could be considered a more sophisticated flow control mechanism. When fully opened, the proposed cylinder-in-ball valve has an even higher *CV* value of 109.87 compared with that of 90.50 for the V-port ball valve.

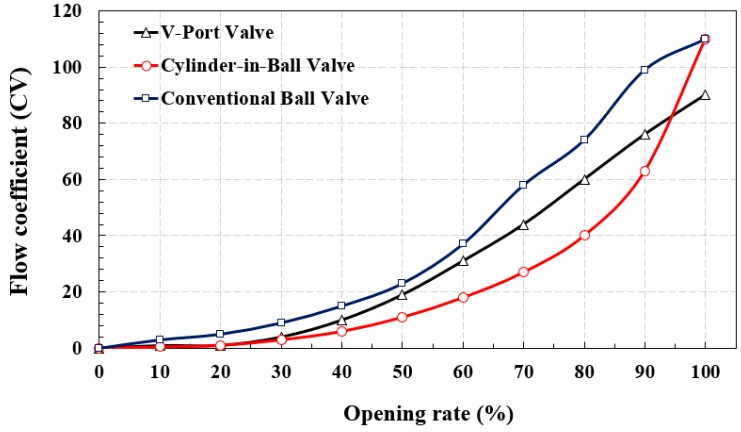

**Figure 6.** The *CV* values to opening rates for the cylinder-in-ball valve, conventional ball valve, and V-port ball valve.

### 4.2. Experiment

To verify the results of the CFD analysis, the *CV* measurement test for the prototype of the proposed cylinder-in-ball valve, as shown in Figure 7, was performed based on the national standard KS B 2101 [14]. The proposed cylinder-in-ball valve was installed in a flow test facility using a closed-loop water piping system, which consisted of a pump and a water reservoir with capacities of 15 bar and 107.25 m$^3$, respectively. Figure 8 shows the experimental equipment and test set-up for the *CV* measurement test. Two digital pressure

gauges and an electromagnetic flowmeter were used during the test to measure pressures and flow rates, respectively. A digital pressure gauge, G1, was installed at the front of the valve to measure the upstream inlet, and the other, G2, was located at the downstream outlet. The electromagnetic flowmeter, EF, was located between G2 and the water reservoir, where the flow rates were subsequently measured once per second, a total of five times. The *CV* measurement test continued to measure the flow rates of the fully opened proposed cylinder-in-ball valve until the $\Delta P$ obtained by G1 and G2 reached 74 kPa. When the test results were plugged into Equation (1), the average flow rate was 72.0 m³/hr, and the *CV* was 97.3. A comparison between the experimental and numerical results of the flow rates and *CV* shows good agreement, as listed in Table 2.

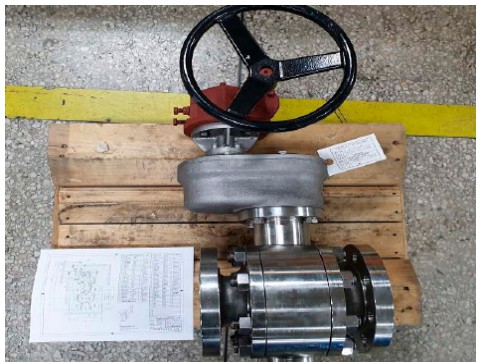

**Figure 7.** Prototype of the proposed cylinder-in-ball valve.

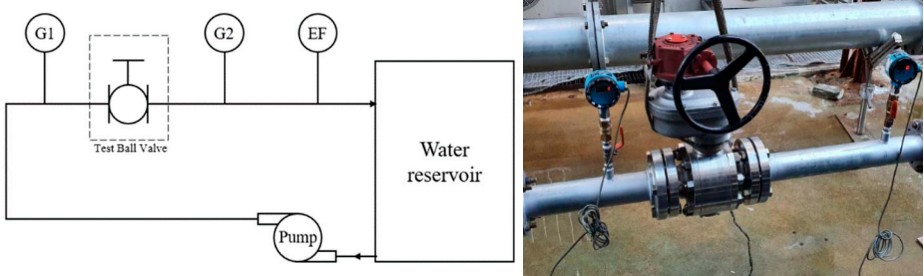

**Figure 8.** Experimental equipment and test set-up for *CV* measurement.

**Table 2.** Technical specifications of flow test facility and results of experimental and numerical analysis for the fully opened proposed cylinder-in-ball valve.

| Technical Specification of Flow Test Facility | |
| --- | --- |
| Max. pressure of pump | 15 bar |
| Capacity of water reservoir | 107.25 m³ |
| Measurement range of digital pressure gauge | 0~1000 kPa |
| Test standard | KS B 2101 |

| Results of experimental and numerical analysis | | |
| --- | --- | --- |
| | Experiment | Analysis |
| Flow rates | 72.0 m³/h | 80.78 m³/h |
| *CV* | 97.3 | 109.87 |

### 4.3. Cavitation

The cavitation of the proposed cylinder-in-ball valve was evaluated quantitatively using the results from the multi-phase CFD analysis. Figure 9 shows the obtained results for the contours of the pressure and the vapor volume fraction, *VVF*, at an opening rate of 20%. The pressure did not change significantly for the entire streamline, where the range of pressure changes was only from 101.05 to 102.77 kPa, as shown in Figure 9a. By

substituting the results obtained from the pressure distribution into the Rayleigh–Plesset equation, the cavity size, such as the radius of the vapors, $R_B$, was calculated, and then the *VVF* was evaluated, as shown in Figure 9b. No meaningful *VVF* was found. In general, the occurrence of cavities was defined as when the *VVF* reached $10^{-5}$, but it was reasonable to regard a level of 0.5 or more, which could cause noticeable damage to the valve [19]. Therefore, cavitation did not occur in the proposed cylinder-in-ball valve even with a small opening rate, and the irregular flow characteristics that commonly occur when using a conventional ball valve were avoided.

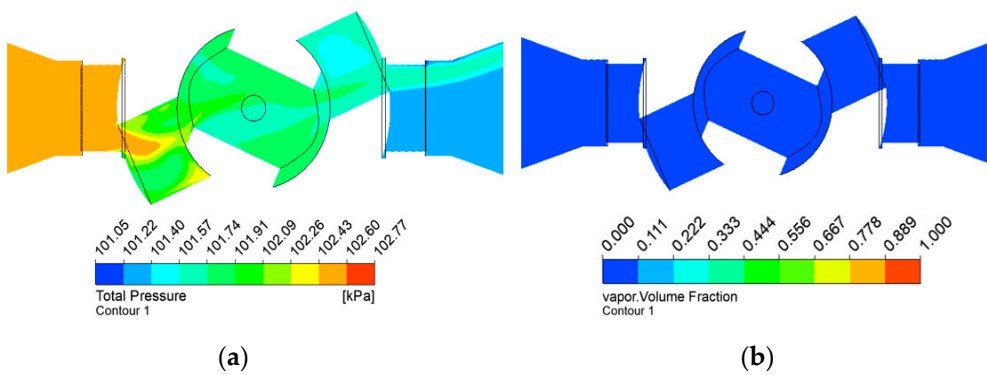

**Figure 9.** Contour plots of the pressure and vapor volume fraction for the proposed cylinder-in-ball valve at an opening rate of 20%: (**a**) pressure; (**b**) vapor volume fraction.

Meanwhile, the cavitation of the V-port ball valve with a V-port angle of 90° was estimated at the same opening rate. Figure 10 shows the contours of the velocity of streamlines and the *VVF*. The valve opening starts from the side and causes the flow path to change rapidly, as shown in Figure 10a. The velocity in the streamlines changed remarkably from 0.168 to 8.657 m/s. This change caused a drastic pressure drop, and the *VVF* reached 1.0, as shown in Figure 10b. The possibility of damage in a piping system due to cavitation is expected.

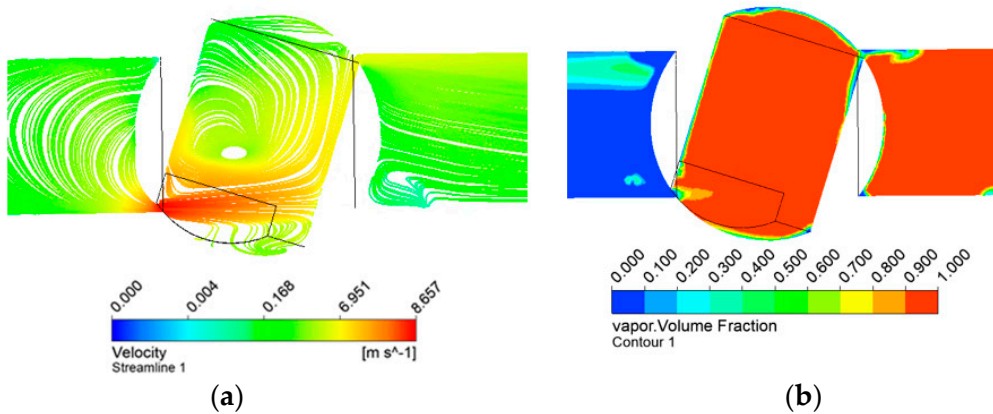

**Figure 10.** Contour plots of the velocity and vapor volume fraction for a V-port ball valve at an opening rate of 20%: (**a**) velocity of streamlines; (**b**) vapor volume fraction.

In this section, quantitative evaluations of the cavitation in the cylinder-in-ball valve and the V-port valve at a small opening rate of 20% were carried out. The coaxial counter-rotating motion of the ball and cylinder in the cylinder-in-ball valve was able to stabilize the flow path in the flow fields and significantly reduce the occurrence of vortices. This sophisticated flow control mechanism improved the irregular flow characteristics that appear frequently in the conventional ball valve. In addition, the cavitation that was one of the drawbacks of the V-port valve was avoided.

## 5. Conclusions

In this study, a cylinder-in-ball valve, in which the opening hole can start at the center of the flow path, was proposed. To quantitatively evaluate the flow characteristics and the cavitation phenomenon, computational fluid dynamic analysis was carried out. The results were reviewed via comparisons with a V-port ball valve. In addition, an experimental *CV* measurement was performed, and the results were compared with those obtained from numerical analysis for the proposed cylinder-in-ball valve. The results are as follows.

(1) Through a structural conceptual design, a cylinder-in-ball valve was proposed, in which the ball and cylinder were rotated in opposite directions using a combination of gears. This created a coaxial counter-rotating motion of the ball and cylinder, and the valve opening started at the center of the flow path.

(2) The estimates for the *CV* of the proposed cylinder-in-ball valve ranged from 1.05 to 109.87 as the opening rate changed from 20 to 100%, and the change in the *CV* for the opening rates followed an even distribution of exponential increments and provided an equal-percentage of inherent flow characteristics for a flow-control valve.

(3) The proposed cylinder-in-ball valve provided an insensitive *CV* change at the range of opening rates between 40 and 80% compared with that of the V-port ball valve, so the proposed cylinder-in-ball valve could be considered to offer a more sophisticated level of flow control.

(4) The experimentally obtained *CV* values for the fully opened proposed cylinder-in-ball valve showed good agreement with estimates via the CFD analysis.

(5) Multi-phase CFD analysis based on the Rayleigh–Plesset equation provided no meaningful *VVF* for the proposed cylinder-in-ball valve at an opening rate of 20%. Therefore, cavitation did not occur in the proposed cylinder-in-ball valve even at a small opening rate, and the irregular flow characteristics were avoided.

**Author Contributions:** All authors contributed to this work by collaboration. Conceptualization, H.-L.K. and S.-H.H.; software and experiment, H.-L.K.; validation, H.-J.P., H.-L.K. and S.-H.H.; formal analysis, H.-J.P., H.-L.K.; writing—original draft preparation, H.-L.K.; writing—review and editing, S.-H.H.; visualization, H.-J.P.; supervision, S.-H.H. All authors have read and agreed to the published version of the manuscript.

**Funding:** This research was funded by the Ministry of Education of the Republic of Korea (2021R1I1A3042151).

**Institutional Review Board Statement:** Not applicable.

**Informed Consent Statement:** Not applicable.

**Data Availability Statement:** Not applicable.

**Acknowledgments:** This research was supported by the Basic Science Research Program through the National Research Foundation of Korea (NRF) (2021R1I1A3042151).

**Conflicts of Interest:** The authors declare no conflict of interest.

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
