# Peer review of "Investigation of the Flow Characteristics for Cylinder-in-Ball Valve Due to a Change in the Opening Rate"

_applsci, doi:10.3390/app12188930_

Round 1
Reviewer 1 Report
In this study, a cylinder-in-ball valve, in which the opening hole can start at the center of the flow path was proposed. To quantitatively evaluate the flow characteristics and the cavitation phenomenon, computational fluid analysis was carried out. The results were reviewed via comparisons with a V-port ball valve. In addition, experimental CV measurement was performed, and the results were compared with those obtained from numerical analysis. Some comments could be considered as follows: 1.The new structural conceptual design should be discussed with existed results. 2.What is main control problem for this complex dynamic system. please discuss it. 3.The authors just propose cylinder-in-ball valve as a flow-control valve, the main contribution should be discussed. 4. How to control the non-linear flow characteristic? 5.References is not enough, more relevant publications could be considered as DOI: 10.1016/j.isatra.2022.02.019, 2022, DOI:10.1109/TIE.2022.3186348,2022.Author Response
Thank you for giving us the opportunity to strengthen our manuscript with your valuable comments and queries.
Please see the attachment.

Reviewer 2 Report
General comments:
In the article titled “Investigation of the Flow Characteristics for Cylinder-in-Ball Valve due to a Change in the Opening Rate” propose a cylinder-in-ball valve through a structural conceptual design, wherein the opening begins at the center of the flow path. Moreover, the flow characteristics and flow rates according to the opening rate via computational fluid dynamics analysis were evaluated also. However, for a better understanding it should be integrated considering the following aspects:
1. Please add the dimension in Figure 1 for easier understanding.
2. Please add more references especially for the last 5 years.
Author Response
Thank you for giving us the opportunity to strengthen our manuscript with your valuable comments and queries.
Please see the attachment.

Reviewer 3 Report
The introduction section is well known in specialized literature for the flow transport phenomenon. There is no new information to research in the manuscript.
The second section has no measurements for the valve, it is just a conceptual description with three views of a generic 1-1/2 valve.
Equations 1 to 4 are well known. What is the scientific value of this research?
The result section agrees with a CV's typical behavior for generic valves.
Author Response

(The authors gave the same response as above.)

Round 2
Reviewer 1 Report
In the present study, a cylinder-in-ball valve was proposed through a structural conceptual design, wherein the opening begins at the center of the flow path.Some comments as follows:
1.The main conrrol characters should be discussed more clearly.
2.What is the advanatages of structural design.
3.Due to complex control issues, what is controller, the advantages should be dicussed existed results such as DOI:10.1109/TIE.2022.3186348 and DOI:10.1109/TASE.2022.3156943.
4.The test results should described with a table.
Reviewer 3 Report
Authors wrote, "Therefore, CFD analysis should be carried out using boundary conditions such as the Q at the inlet and atmospheric pressure at the outlet, which meets the requirement of ∆P with 74 kPa". This is not enough scientific justification for novelty.
Round 3
Reviewer 3 Report
It a case of study.